# Drivers with and without Obesity Respond Differently to a Multi-Component Health Intervention in Heavy Goods Vehicle Drivers

**DOI:** 10.3390/ijerph192315546

**Published:** 2022-11-23

**Authors:** Katharina Ruettger, Stacy A. Clemes, Yu-Ling Chen, Charlotte L. Edwardson, Amber Guest, Nicholas D. Gilson, Laura J. Gray, Vicki Johnson, Nicola J. Paine, Aron P. Sherry, Mohsen Sayyah, Jacqui Troughton, Veronica Varela-Mato, Thomas Yates, James A. King

**Affiliations:** 1School of Sport, Exercise and Health Sciences, Loughborough University, Loughborough LE11 3TU, UK; 2NIHR Leicester Biomedical Research Centre, Leicester LE5 4PW, UK; 3Diabetes Research Centre, University of Leicester, Leicester General Hospital, Leicester LE5 4PW, UK; 4School of Human Movement and Nutrition Sciences, The University of Queensland, Brisbane 4072, Australia; 5Department of Health Sciences, University of Leicester, Leicester LE1 7RH, UK; 6Leicester Diabetes Centre, University Hospitals of Leicester NHS Trust, Leicester General Hospital, Leicester LE5 4PW, UK

**Keywords:** body weight, occupational health, occupational drivers, physical activity, sedentary behaviour, workplace

## Abstract

Physical inactivity and obesity are widely prevalent in Heavy Goods Vehicle (HGV) drivers. We analysed whether obesity classification influenced the effectiveness of a bespoke structured lifestyle intervention (‘SHIFT’) for HGV drivers. The SHIFT programme was evaluated within a cluster randomised controlled trial, across 25 transport depots in the UK. After baseline assessments, participants within intervention sites received a 6-month multi-component health behaviour change intervention. Intervention responses (verses control) were stratified by obesity status (BMI < 30 kg/m^2^, n = 131; BMI ≥ 30 kg/m^2^, n = 113) and compared using generalised estimating equations. At 6-months, favourable differences were found in daily steps (adjusted mean difference 1827 steps/day, *p* < 0.001) and sedentary time (adjusted mean difference −57 min/day, *p* < 0.001) in drivers with obesity undertaking the intervention, relative to controls with obesity. Similarly, in drivers with obesity, the intervention reduced body weight (adjusted mean difference −2.37 kg, *p* = 0.002) and led to other favourable anthropometric outcomes, verses controls with obesity. Intervention effects were absent for drivers without obesity, and for all drivers at 16–18-months follow-up. Obesity classification influenced HGV drivers’ behavioural responses to a multi-component health-behaviour change intervention. Therefore, the most at-risk commercial drivers appear receptive to a health promotion programme.

## 1. Introduction

Heavy goods vehicle (HGV) drivers face many barriers to adopting healthy lifestyles [1]. Within their working environment, drivers often experience long and variable working hours, pressurised delivery schedules and enforced sedentarism [1]. These barriers to adopting a healthy lifestyle have been exacerbated by the COVID-19 pandemic which has disrupted international supply chains. A notable shortfall in HGV drivers, particularly within the UK (~100,000), has added to the pressures experienced by this occupational group [2]. 

Previous data have shown that HGV drivers have high levels of physical inactivity and sedentary time [3,4], contributing to high levels of overweight and obesity [4,5,6,7]. In many, this phenotype overlaps with sleep deprivation [8] and poor dietary choices, which collectively augment the risk of chronic metabolic disease, a reduced life expectancy, and heighten the risk of road traffic accidents [9,10,11,12,13]. Despite these issues, HGV drivers remain an underserved population with regards to health promotion interventions. 

We therefore designed the Structured Health Intervention for Truckers (SHIFT), a multi-component, theory-driven, health behaviour change intervention seeking to increase physical activity, reduce sitting time and enhance cardiometabolic health in this occupational group [14]. SHIFT has recently been evaluated using a cluster randomised controlled trial (RCT), with 25 transport depots in the UK [15]. Baseline data from the RCT showed that 88% of drivers had overweight or obesity, a prevalence higher than that seen in European drivers [4,16,17] and within the general UK (age-matched) population [18]. At 6-months, a favourable difference in physical activity was apparent between study groups, but this was primarily driven by a decrease in physical activity within the control group. The intervention had no impact on cardiometabolic or anthropometric outcomes [15].

Within the SHIFT cluster RCT, responses in the intervention group were compared to controls without any stratification of study sub-groups. However, previous behaviour change interventions have demonstrated that obesity classification may influence intervention responses [19,20]. Notably, the Health Belief Model suggests that behaviour change is influenced by an individual’s feelings about the seriousness of contracting an illness or disease, and its medical consequences [21,22,23]. In the SHIFT trial, all participants received a detailed health assessment at baseline and were provided with feedback, including information about their future health risk. Based on this information, we hypothesised that obesity may influence physical activity, anthropometric and cardiometabolic health responses to the SHIFT programme. Specifically, we anticipated that more potent intervention responses would be seen in drivers living with obesity due to them having a more extensive behaviour change, after having recognised that their body mass index (BMI) places them at a greater risk of several long-term health conditions and premature mortality. 

Therefore, this secondary data analysis investigated whether BMI influenced the effectiveness of responses to the SHIFT programme. With 47% of our sample possessing a BMI of ≥30 kg/m^2^ at baseline, we were able to re-analyse data after segregating responses to intervention and control arms based on obesity classification. Our primary analyses were based on responses at 6-months follow-up; however, the potential longer-term impact of the intervention was assessed for some outcome measures at 16–18-months follow-up.

## 2. Materials and Methods

### 2.1. Study Design, Setting and Participants

This study used data collected from participants taking part in the wider SHIFT RCT [15]. The full protocol of this cluster RCT, and the main findings of the effectiveness trial, have been published elsewhere [14,15]. Measurements and procedures relevant to the present analyses are described below. 

Participants were full-time long-distance HGV drivers (drivers who cover long distances with few delivery stops), recruited from an international logistics company across 25 participating depots throughout the Midlands region in the UK. Drivers were eligible to participate if they had no clinically diagnosed cardiovascular disease, haemophilia, blood-borne viruses or mobility limitations. Institutional ethical committee approval was obtained before study procedures commenced. All participants provided written informed consent before baseline assessments and were re-consented before follow-up measures.

### 2.2. Measurements

Baseline data were collected between January 2018 and July 2019 within a 2-h assessment. Follow-up data were collected 6-months after the baseline assessment and included the same measurements. Due to the COVID-19 pandemic, measures at 16–18 months follow-up were limited to device-assessed physical activity and self-reported data, as face-to-face data collection was not permitted.

The assessments took place within the worksite setting and were undertaken by research staff trained in the study’s Standard Operating Procedures. Participants fasted for at least 4-h before assessments, which occurred at the beginning or end of drivers’ shifts. Participants reported basic demographic information via questionnaires. Drivers also provided information about their current health status, use of medication and smoking status. Diet was measured using a short form food frequency questionnaire [24]. A dietary quality score (DQS) was then calculated based on fruit, vegetable, oily fish, fat, and non-milk extrinsic sugar intake. Scores between one to three were given for each category, with a score of three indicating that the UK recommendations were met for that component. The maximum achievable DQS was 15 [24].

Bio-electrical-impedance scales (DC-360S, Tanita Corporation, Tokyo, Japan) were used to measure body mass and body fat percentage. Height was assessed using a portable stadiometer (Seca 206, Oxford, UK). Circumferences (waist, hip, neck) were measured according to World Health Organisation (WHO) guidelines [25]. Resting blood pressure (BP) and heart rate were measured using an automated sphygmomanometer (Omron HEM-907, Omron Corporation, Kyoto, Japan) according to the European Society of Hypertension guidelines [26]. Normotensive BP, pre-hypertension and hypertension were classified as recommended by Williams et al. [27]. A Takei Hand-Grip dynamometer (Takei Scientific Instruments Co., Ltd., Niigata City, Japan) was used to measure grip strength from both hands. Glycated haemoglobin (A1CNow^®+^, PTS Diagnostics, Indianapolis, USA), triacylglycerol, high-density lipoproteins (HDL), and total cholesterol (Cardiocheck^®®+^, PTS Diagnostics, Indianapolis, USA) were measured from finger-prick blood samples. The Friedewald formula [28] was used to calculate low-density lipoprotein cholesterol (LDL). The risk of having a cardiovascular event within the next 10 years was calculated using the QRISK2 algorithm [29].

### 2.3. Physical Activity and Sitting Time

activPAL3 micro accelerometers (PAL Technologies Ltd., Glasgow, UK) were used to measure daily step counts, sitting time, and time spent in light physical activity (LPA, for example, slow walking) and moderate-to-vigorous physical activity (MVPA, for example, brisk walking through running, or any activity which increases heart rate, respiration and body temperature). The monitor was placed on participants’ non-dominant side on the midline anterior aspect of the upper thigh. The device was waterproofed to enable participants to wear it continuously for eight consecutive days after each assessment. Additionally, participants were asked to record their waking, sleep, working and non-wear times via daily logs. Data from the activPAL were downloaded using manufacturer proprietary software (activPAL Professional v.7.2.38, Glasgow, Scotland, UK). The generated event files were processed using the Processing PAL software (https://github.com/UOL-COLS/ProcessingPAL, version 1.3, University of Leicester, (Leicester, UK)). The full protocol for the data processing has been described elsewhere [15,30]. Participants had to provide at least one valid day of data at both time points to be included in the physical activity analysis of this study. Participants were included in this sub-analysis for work and non-workdays if they provided valid data on at least one working day and one non-working day at both time points. A valid day was defined as: >10 h of valid waking wear time, >500 single leg steps (i.e., >1000 steps/day), and <95% of time spent in any one behaviour (e.g., sitting, standing or stepping) [31]. Key variables that were extracted from the data were number of steps per day, time spent standing, stepping and sitting (total and long sitting bouts (>30 min)), the number of sitting-to-upright transitions and the amount of time spent in LPA and MVPA. MVPA was defined as a step cadence of ≥100 steps per minute which was accumulated in bouts of at least one minute. LPA was calculated by subtracting MVPA, sitting and standing time from the valid waking wear time.

### 2.4. The Structured Health Intervention for Truckers (SHIFT) 

After the baseline assessment, participating sites were randomised (1:1) into intervention or control groups. The 6-month intervention consisted of a group-based (4–6 participants) 6-h structured education session tailored for HGV drivers, underpinned by the Social Cognitive Theory [32]. The session was delivered by 2 trained Educators (a member of the research team in collaboration with personnel from the logistics organisation) and took place within appropriate training rooms within the intervention depots. Participants were supported to work out knowledge through group discussions instead of being taught in a didactic way. The discussions and interactive activities included strategies for participants to increase their physical activity, improve their diet and reduce their sitting time in and outside of work [14]. Risk factors for type 2 diabetes and cardiovascular disease were also discussed. Participants received a Fitbit^®^ Charge 2 (Fitbit, Inc., San Francisco, CA, USA) activity tracker within the session which provides feedback on their daily step count during their weekly routines. The drivers were encouraged to use this device to set goals to gradually increase their physical activity through walking-based activity [14]. Additionally, participants were able to share their daily activity with two members of the research team via an online monitoring system (Fitabase, Small Steps Labs LLC, San Diego, CA, USA, https://fitabase.com/). These data were used to provide individually tailored step count challenges every 6-weeks for each participant throughout the intervention period. Participants received exercise equipment (resistance bands, a ball, and a grip strength dynamometer) for use in their cab, when not permitted to leave the vehicle or whilst taking a break. The Educators introduced and practised the “cab workout” with drivers within the education session involving light-intensity resistance exercises. Personnel from the company co-delivering the education sessions also acted as local champions and provided ongoing health coach support, along with members of the research team via a text messaging service [15]. 

All participants from the control and intervention arms undertook the same measurements and received feedback immediately following their assessments at baseline and 6-months follow-up. Participants randomised into the control arm received a leaflet containing basic general guidance on physical activity, sleep and diet, but otherwise continued with their usual practice throughout the 6-month intervention period.

### 2.5. Data Analyses

All variables were checked for normality using Kolmogorov Smirnov tests along with histograms. Normally distributed data at baseline were reported as means (standard deviation [SD]), non-normally distributed data reported as medians (interquartile range [IQR]), and frequencies and proportions (%) were reported for categorical variables. Changes from baseline are presented as mean (SD). In separate analyses, participants were divided into two groups based on their BMI (<30 kg/m^2^ and ≥30 kg/m^2^).

Outcome variables were analysed using generalised estimating equations (GEE), taking account of clustering by depot centres. Given the focus of the paper was specifically on obesity, the models were run separately a priori for drivers with obesity and drivers without obesity, with outcomes compared between the intervention and control arms. Models were adjusted for each variable at baseline and the cluster size category (small < 40 drivers/site; large ≥ 40 drivers/site). Physical activity outcomes were additionally adjusted for change in valid waking wear time from baseline to 6-months follow-up. The intervention effect is presented as beta coefficients (95% CI), showing the adjusted difference between intervention and control groups. *p* values of <0.05 were considered as significant. Adjustment for multiple testing was not undertaken, therefore results are interpreted with caution in relation to the overall pattern of findings. The above analyses were repeated using waist circumference (<102 cm and ≥102 cm for men and <88 cm and ≥88 cm for women) and body fat percentage (≤25% and ≥25% for men and <35% and ≥35% for women) categories as sensitivity analysis to show if the results could be confirmed classifying obesity with different measurements. Statistical analyses were conducted using IBM SPSS V.27 (IBM, Armonk, NY, USA).

## 3. Results

A total of 382 participants (98.7% male) from 25 sites were recruited into the main trial. Two hundred and forty-four participants provided cardiometabolic health data at baseline and 6-months follow-up, and were included in the analyses reported herein. Thirteen sites were randomly assigned to the control arm (142 participants), and 12 sites to the intervention arm (102 participants). A CONSORT diagram describing participant numbers throughout the study has been reported elsewhere [15]. Table 1 shows the sociodemographic information, medical data, cardiometabolic biomarker and lifestyle behaviours within trial arms for participants with (N = 113) and without obesity (N = 131) at baseline categorised by BMI (≤30 or ≥30 kg/m^2^). 

### 3.1. Physical Activity

Table 2 details physical activity and sedentary time data recorded across all valid days (N = 207). At 6-months, participants with obesity (based on BMI) from the intervention arm accumulated more daily steps (1827 steps/day) and had a lower sitting time (−57 min/day), than those with obesity in the control arm. There were no significant intervention effects in the group without obesity. Appendix A present physical activity and sedentary time data measured on workdays and non-workdays (N = 171). On workdays, in participants with obesity, there was an intervention effect for all outcomes besides sit-to-stand transitions and time spent in MVPA, relative to controls with obesity. Conversely, in the intervention group without obesity, no intervention effects were apparent on workdays. On non-workdays, improvements in all movement-related outcomes except sit-to-stand transitions were observed for participants with obesity from the intervention group, relative to controls. In the intervention group without obesity, the only effect identified was a reduction in MVPA to a lower magnitude than that seen in the control group without obesity. These results were mirrored in sensitivity analyses when obesity status was additionally defined based on body fat percentage or waist circumference (Appendix A).

### 3.2. Cardiometabolic and Lifestyle Outcomes

In participants with obesity, the SHIFT programme improved several cardiometabolic biomarkers, anthropometric and lifestyle outcomes, relative to controls with obesity, which were not observed in the SHIFT intervention group without obesity (Table 3). Specifically, in comparison to the control group with obesity, body weight, BMI, waist circumference and neck circumference were reduced in the intervention group with obesity; whilst HDL was increased. Conversely, a reduction in resting heart rate was the only intervention effect in participants without obesity (verses the non-obese control group). These results are consistent when obesity stratification is additionally made based on either body fat percentage or waist circumference (Appendix A).

### 3.3. Extended Follow-Up

Appendix A show physical activity and dietary outcomes 16–18 months post-randomisation. Two hundred and twelve participants provided self-reported data, and 163 participants additionally provided valid physical activity data. No significant differences between intervention and control groups were apparent in participants with or without obesity.

## 4. Discussion

This secondary analysis investigated whether obesity classification at baseline influenced the effectiveness of the SHIFT programme. With regards to physical activity, sitting time and anthropometric outcomes, this secondary analysis has shown that favourable responses to the SHIFT programme were only seen in drivers with obesity, suggesting different engagement and response in this group. These data have important implications concerning the implementation of the SHIFT programme within the transport and logistics sector and highlight the necessity of the specific needs and characteristics of HGV drivers.

When specifically examining drivers with obesity, our analyses showed that the intervention effect at 6-months equated to a difference in physical activity of 1827 steps/day, relative to controls with obesity. This was generated by an increase in steps (774 steps/day) in the intervention group and a decrease in the control group (−1129 steps/day). This physical activity response was superior to that observed in the SHIFT main outcomes trial, where data from participants with and without obesity were reported collectively [15]. Specifically, in the RCT, the difference reported between intervention and control arms at 6-months was driven primarily by a reduction in steps in the control group (−716 steps/day), rather than an increase in the intervention group (+32 steps/day). Importantly, in the present analyses, the SHIFT programme did not alter physical activity levels in the intervention group without obesity. As outlined within behaviour change theory [21,22,23], this divergent response may reflect greater receptiveness to the health messages within the SHIFT intervention, as those with obesity recognise their elevated health risk. This scenario may have been particularly likely in the present study, as all participants (irrespective of group assignment) were provided with detailed feedback about their health status, immediately after baseline assessments. This impact of obesity classification on physical activity responses to a health intervention has been reported previously [19]. 

The increase in daily steps in drivers with obesity from the intervention group was 4-fold greater on non-workdays in comparison to workdays. This finding likely reflects the enforced sedentariness imposed by drivers’ working days [1] and the greater ease of changing behaviour during leisure time. However, it was encouraging that drivers with obesity from the intervention arm still managed to reduce their sitting time on workdays by 33 min/day. Previous research has shown that reallocating 30 min of sedentary time per day to LPA is associated with subtle improvements in cardiometabolic health markers [33]. Once more, however, the intervention did not elicit this response in drivers without obesity; in fact, their sitting time was higher at 6-months on workdays and non-workdays, whilst their daily steps were reduced. Collectively, this improved ‘movement profile’, catalysed by the SHIFT programme in drivers with obesity, is likely to benefit their health and wellness. Indeed, recent research indicates that an increase in daily steps of 500 per day is the minimum necessary to improve long-term health and mortality risk in inactive populations, with greater benefits achievable for those able to do more [34].

The longevity of responses elicited by health interventions is always crucial. Unfortunately, after the 6-month assessments, the SHIFT RCT was impacted by the COVID-19 pandemic. Whilst we were able to deploy accelerometers at 16–18 months follow-up, data were only available for a sub-sample of participants, and it is likely that the pandemic will have impacted drivers’ behaviour. Our present analyses identified no differences in physical activity or sitting time in any group at this extended follow-up; implying that the improvements seen at 6-months in the intervention group with obesity had not been maintained. Additional work is needed to develop strategies which ensure that positive behaviour change is supported over the longer term.

The present analyses also identified a more potent intervention effect on body weight in participants with obesity, compared with the overall effect in the SHIFT RCT [15]. Intervention participants with obesity lost 2.5 kg (2.4%) of body weight at 6-months, which was eight-fold greater than that seen in the control group with obesity. The increase in steps observed in drivers with obesity at 6 months is unlikely to fully explain the weight loss observed in this group alone, suggesting that some changes in dietary behaviours may have also occurred. Whilst dietary changes were reported by intervention participants within our process evaluation [35], the sensitivity of our Food Frequency Questionnaire was not sufficient however to detect changes in dietary behaviours [15].

Notably, the SHIFT intervention did not impact body weight in the intervention group without obesity. This divergent response, between participants with and without obesity, may have been partially expected, given the former group had greater adiposity at baseline. However, it should be recognised that the group without obesity still possessed a mean BMI in the overweight range, with mildly elevated body fat. The intervention findings in drivers with obesity are encouraging given the high prevalence of obesity in HGV drivers [5,6,7,36], particularly as obesity has been linked to an increase in road traffic accident risk in previous studies [9]. Additionally, from an individual health perspective, the concomitant reduction in waist and neck circumference in the intervention group with obesity, is particularly notable given the link between adiposity in these locations and cardiometabolic health risk [37,38]. Relatedly, a subtle increase in HDL was apparent in the intervention group with obesity in our analyses; however, the limited sensitivity of point-of-care blood biomarker analysis used in the present study, may have constrained our ability to detect additional biomarker improvements.

Several weight-loss interventions have previously been conducted on truck drivers [39]. However, before the SHIFT cluster RCT, just two RCTs had been undertaken [5,40] with only one non-RCT demonstrating long-term effectiveness for reducing body weight [41]. One of the RCTs showed similar weight loss outcomes to the present study (−3.31 kg weight reduction at 6-months) [5]. However, weight loss was the primary outcome in this previous study and the intervention components were more intensive. Within this context, the weight-related findings outlined in the present study are encouraging and suggest that the SHIFT programme may facilitate initial weight loss efforts in HGV drivers with obesity. It should be noted, however, that 5–10% weight loss is recommended to provide therapeutic benefits for many obesity-related conditions [42,43]. Therefore, further refinement of the SHIFT programme is needed to support more clinically relevant weight loss.

The secondary analyses reported within this study were enabled by the large and diverse sample recruited within the SHIFT cluster RCT [15]. Although the detailed assessments conducted at baseline and 6-months are a key strength of the SHIFT RCT, it should be recognised that the use of point-of-care blood biomarker analysers may have constrained the ability to detect intervention effects on cardiometabolic biomarkers. The lack of long-term follow-up data must also be considered when interpreting these analyses. As we were not powered for these sub-group analyses, the findings should be interpreted as preliminary.

## 5. Conclusions

In conclusion, this secondary analysis of the SHIFT cluster RCT has demonstrated that obesity classification influencedintervention effectiveness. Specifically, in drivers with obesity, the SHIFT programme beneficially changed daily physical activity, sitting time and body weight. These intervention effects were not mirrored in drivers without obesity. These data highlight the importance of personalising health interventions within this occupational group. Additional research is now needed to further develop the SHIFT intervention into an industry accredited Continued Professional Competency (CPC) module which can be accessible across the UK.

## Figures and Tables

**Table 1 ijerph-19-15546-t001:** Baseline characteristics for participants with versus without obesity based on BMI.

	Participants without Obesity Based on Baseline BMI (BMI < 30 kg/m^2^) Median (IQR), Mean (SD) or % N = 131		Participants with Obesity Based on Baseline BMI (BMI ≥ 30 kg/m^2^) Median (IQR), Mean (SD) or % N = 113	
Demographics	Intervention N = 51	Control N = 80	Intervention N = 51	Control N = 62
Age (years) *	50 (55, 43)	49 (55, 39)	48 (54, 39)	47 (55, 41)
Average working hours/week *	48 (50, 43)	48 (50, 45)	48 (50, 45)	48 (50, 45)
Ethnicity (%) White European Other	96.2 3.8	91.4 8.6	96.1 3.9	95.1 4.9
Highest level of education (%) GCSEs A-Level University Other	54.9 13.7 7.9 23.5	56.3 8.8 10.0 24.9	68.6 7.8 5.9 17.7	59.7 8.1 8.1 24.1
**Medical information (%)**				
Cholesterol medication	9.8	2.5	9.8	12.9
Blood pressure medication	9.8	8.8	13.7	14.5
Diabetes medication	2.0	2.5	3.9	8.1
Other medication	15.7	22.5	29.4	22.6
Q-Risk (%) Less than 10% 10% or over 20% or over	80.4 19.6	80.0 13.8 6.3	72.5 23.5 3.9	72.6 24.2 3.2
**Anthropometric measures**				
Body fat %	23.6 (4.3)	22.8 (4.6)	30.7 (4.3)	31.0 (4.1)
Weight (kg) *	85.1 (90.1, 80.2)	84.6 (92.6, 76.2)	105.0 (117.7, 97.8)	108.3 (117.2, 98.7)
BMI (kg/m^2^) *	27.0 (28.5, 26.1)	27.2 (28.6, 24.9)	33.2 (35.8, 31.1)	33.3 (36.0, 31.1)
Waist Circumference (cm) *	96.0 (101.0, 92.0)	95.4 (100.9, 90.2)	112.0 (120.0, 104.0)	113.7 (122.0, 106.9)
Hip Circumference (cm) *	104.0 (107.0, 101.0)	101.6 (105.8, 97.6)	113.0 (118.5, 108.5)	112.0 (118.1, 107.2)
Neck Circumference (cm) *	39.0 (41.0, 37.0)	39.2 (40.1, 37.4)	42.0 (44.1, 40.3)	42.0 (44.0, 40.2)
Grip strength (kg) *	50.5 (54.5, 44.3)	50.0 (55.9, 43.4)	52.0 (58.5, 46.5)	52.0 (57.8, 46.4)
**Blood pressure**				
Systolic Blood pressure (mm Hg) *	130.0 (137.5, 118.5)	125.3 (134.3, 118.8)	130.0 (138.5, 122.0)	133.0 (144.5, 124.8)
Diastolic Blood pressure (mm Hg)	80.8 (9.1)	79.5 (10.6)	83.3 (8.7)	85.4 (9.5)
Resting heart rate (beats/min)	64.9 (10.1)	65.7 (9.9)	69.7 (11.3)	69.0 (9.3)
**Blood markers**				
HbA1c (mmol/mol) *	33 (36, 30)	35 (37, 32)	35 (38, 32)	37 (39, 34)
Triglycerides (mmol/L) *	1.23 (1.78, 0.94)	1.13 (1.80, 0.81)	1.56 (2.40, 1.03)	1.41 (2.25, 1.08)
HDL Cholesterol (mmol/L) *	1.20 (1.53, 1.05)	1.21 (1.41, 1.00)	1.07 (1.27, 0.93)	1.11 (1.35, 0.90)
LDL Cholesterol (mmol/L)	2.80 (0.86)	2.89 (0.91)	2.80 (0.72)	2.84 (0.75)
Total Cholesterol (mmol/L)	4.38 (0.96)	4.42 (0.97)	4.29 (0.88)	4.41 (0.90)
**Lifestyle behaviours**				
Alcohol units/week *	9.0 (20.0, 2.0)	4.0 (9.8, 1.5)	5.5 (10.0, 1.5)	6.8 (14.0, 1.5)
Current Smoker (%)	3.8	9.9	7.1	11.5
Fruit intake (grams/day) *	56.8 (280.0, 28.8)	56.8 (120.0, 28.8)	56.8 (120.0, 11.2)	56.8 (120.0, 24.4)
Vegetable intake (grams/day) *	85.6 (176.8, 57.6)	68.0 (113.6, 40.0)	68.0 (113.6, 57.6)	60.8 (113.6, 40.0)
Dietary Quality Score *	11.0 (13.0, 10.0)	12.0 (13.0, 10.0)	11.0 (13.0, 9.0)	11.0 (12.0, 10.0)
**Physical activity and sitting behaviours**				
Waking wear time (min/day) *	995.3 (962.6, 1039.5)	985.1 (950.3, 1027.6)	995.7 (960.86, 1033.37)	1009.4 (961.19, 1036.47)
Steps/day *	10,070 (7661, 12,664)	8527 (6474, 10,510)	7751 (6554, 9800)	8770 (7028, 10,138)
Time spent sitting (min/day)	648.65 (13.29)	664.7 (12.25)	680.2 (14.16)	688.6 (11.35)
Sitting bouts > 30 min (min/day) *	400.2 (269.1, 456.9)	434.05 (376.5, 498.5)	421.8 (324.5, 500.3)	435.6 (385.5, 510.5)
Time spent standing (min/day) *	215.8 (185.7, 248.9)	194.7 (165.6, 249.7)	197.1 (169.4, 232.0)	197.4 (166.6, 232.0)
Time spent stepping (min/day) *	128.0 (104.2, 154.7)	112.5 (86.2, 140.0)	100.9 (87.2, 131.3)	117.1 (90.7, 133.8)
Sit to upright transitions (n) *	51.3 (43.4, 64.1)	49.7 (40.5, 57.8)	45.9 (39.2, 53.3)	44.1 (37.3, 55.3)
Time spent in MVPA (min/day) *	14.6 (6.9, 25.5)	10.5 (7.1, 18.3)	8.9 (4.5, 13.0)	9.3 (6.9, 15.9)
Time spent in LPA (min/day) *	105.3 (87.4, 137.3)	95.9 (78.7, 116.4)	93.2 (77.6, 119.1)	99.3 (81.4, 113.3)

Note: * Data are presented using the median and inter-quartile range. Abbreviations: BMI = Body Mass Index; GSCE = General Certificate of Secondary education; HbA1c = Haemoglobin A1c; HDL = High-density lipoprotein; IQR = Interquartile Range; LDL = Low-density lipoprotein; LPA = Light physical activity; MVPA = Moderate-to-vigorous physical activity; SD = Standard deviation.

**Table 2 ijerph-19-15546-t002:** Physical activity and sitting behaviours measured across all valid days for participants with and without obesity based on BMI.

	Participants without Obesity Based on Baseline BMI (BMI < 30 kg/m^2^) N = 112				Participants with Obesity Based on Baseline BMI (BMI ≥ 30 kg/m^2^) N = 95			
Physical Activity Marker Overall	Change from Baseline (Mean (SD))		Intervention Effect * (95% CI)	*p*-Value	Change from Baseline (Mean (SD))		Intervention Effect * (95% CI)	*p*-Value
	Intervention N = 45	Control N = 67			Intervention N = 44	Control N = 51		
Steps/day	−570 (2768)	−416 (2130)	132.69 (−721.95, 987.32)	0.761	774 (2893)	−1129 (2048)	1827.01 (967.77, 2686.24)	**<0.001**
Time spent sitting (min/day)	6.16 (63.73)	14.31 (84.31)	4.71 (−13.32, 22.74)	0.609	−26.58 (85.67)	30.05 (70.78)	−57.04 (−80.25, −33.83)	**<0.001**
Sitting bouts > 30 min (min/day)	16.05 (81.11)	15.55 (100.70)	1.45 (−26.37, 29.28)	0.918	−23.93 (92.93)	38.53 (78.71)	−69.03 (−97.11, −40.95)	**<0.001**
Time spent standing (min/day)	−5.33 (29.71)	−0.45 (36.09)	−3.57 (−16.40, 9.27)	0.585	8.73 (60.66)	−22.68 (34.24)	32.75 (14.36, 51.13)	**<0.001**
Time spent stepping (min/day)	−8.18 (29.38)	−4.89 (21.73)	−0.20 (−9.47, 9.07)	0.966	8.99 (30.48)	−13.29 (23.72)	22.10 (12.48, 31.72)	**<0.001**
Sitting–to–being upright transitions (n)	−1.38 (14.87)	−0.58 (11.25)	−0.09 (−5.16, 4.97)	0.971	0.36 (12.03)	−2.85 (13.15)	3.72 (−0.96, 8.39)	0.119
Time spent in MVPA (min/day)	1.23 (19.73)	−1.67 (14.88)	4.12 (−1.49, 9.74)	0.150	3.98 (18.26)	−3.37 (13.36)	6.65 (1.61, 11.69)	**0.010**
Time spent in LPA (min/day)	−9.43 (24.25)	−3.22 (17.02)	−3.34 (−11.21, 4.52)	0.405	5.01 (21.78)	−9.92 (20.43)	15.26 (7.47, 23.05)	**<0.001**

Note: * Change in intervention relative to control adjusted for variables at baseline, and change in valid waking wear time from baseline to 6-month follow-up and cluster size category (Small < 40; Large ≥ 40). Abbreviations: CI = Confidence Interval; LPA = Light physical activity; MVPA = Moderate-to-vigorous physical activity; SD = Standard Deviation.

**Table 3 ijerph-19-15546-t003:** Cardiometabolic and lifestyle secondary outcome changes from baseline to 6-month follow-up in participants with and without obesity (based on BMI).

	Participants without Obesity Based on Baseline BMI (BMI < 30 kg/m^2^) N = 131				Participants with Obesity Based on Baseline BMI (BMI ≥ 30 kg/m^2^) N = 113			
Anthropometric Measures	Change from Baseline (Mean (SD))		Intervention Effect * (95% CI)	*p*-Value	Change from Baseline (Mean (SD))		Intervention Effect * (95% CI)	*p*-Value
	Intervention N = 51	Control N = 80			Intervention N = 51	Control N = 62		
Body fat (%)	0.09 (2.22)	−0.03 (2.29)	0.26 (−0.45, 0.97)	0.476	−0.61 (2.22)	0.02 (1.57)	−0.66 (−1.39, 0.06)	0.071
Weight (kg)	−0.50 (4.11)	−0.08 (4.10)	−0.41 (−1.71, 0.89)	0.535	−2.51 (5.95)	−0.29 (5.26)	−2.37 (−4.39, −0.34)	**0.022**
BMI (kg/m^2^)	−0.04 (1.19)	−0.01 (1.11)	−0.02 (−0.39, 0.35)	0.933	−0.71 (1.85)	−0.04 (1.70)	−0.70 (−1.35, −0.06)	**0.032**
Waist Circumference (cm)	−0.61 (5.64)	−0.31 (4.92)	−0.31 (−2.11, 1.49)	0.735	−2.09 (7.30)	0.38 (5.81)	−2.47 (−4.88, −0.05)	**0.045**
Hip Circumference (cm)	−0.50 (3.22)	0.13 (3.76)	−0.35 (−1.49, 0.79)	0.545	−1.39 (4.25)	−0.26 (5.31)	−1.25 (−2.89, 0.39)	0.134
Neck Circumference (cm)	−0.15 (1.55)	0.34 (1.74)	−0.40 (−0.96, 0.16)	0.164	−0.32 (2.15)	0.58 (1.84)	−0.89 (−1.59, −0.20)	**0.011**
Grip strength (kg)	0.64 (6.11)	0.53 (5.61)	0.01 (−2.03, 2.05)	0.992	0.86 (4.78)	−0.46 (5.72)	1.34 (−0.51, 3.19)	0.155
**Blood pressure**								
Systolic Blood pressure (mm Hg)	−2.54 (11.01)	−2.80 (10.95)	1.10 (−2.44, 4.64)	0.543	−2.60 (10.21)	−1.41 (14.89)	−2.14 (−6.55, 2.28)	0.343
Diastolic Blood pressure (mm Hg)	−0.39 (8.90)	−0.76 (8.16)	0.68 (−1.93, 3.28)	0.61	−1.81 (7.16)	−0.72 (9.22)	−1.52 (−4.41, 1.37)	0.303
Resting heart rate (beats/min)	−1.73 (9.99)	1.09 (9.53)	−3.04 (−5.89, −0.19)	**0.037**	−2.27 (10.08)	−2.25 (8.17)	0.29 (−2.71, 3.29)	0.851
**Blood markers**								
HbA1c (mmol/mol)	0.31 (4.91)	0.54 (5.51)	−1.20 (−2.72, 0.31)	0.119	−1.14 (8.15)	0.34 (6.74)	−1.62 (−4.28, 1.04)	0.233
Triglycerides (mmol/L)	0.03 (0.72)	0.05 (1.12)	−0.06 (−0.322, 0.199)	0.643	0.06 (1.04)	0.05 (0.79)	−0.017 (−0.342, 0.309)	0.920
HDL Cholesterol (mmol/L)	0.05 (0.25)	0.04 (0.25)	0.04 (−0.03, 0.12)	0.237	0.10 (0.25)	−0.01 (0.25)	0.08 (0.01, 0.15)	**0.020**
LDL Cholesterol (mmol/L)	−0.06 (0.77)	−0.09 (0.94)	−0.11 (−0.39, 0.16)	0.416	0.06 (0.82)	0.11 (0.73)	−0.06 (−0.03, 0.21)	0.648
Total Cholesterol (mmol/L)	0.01 (0.86)	0.01 (0.98)	−0.09 (−0.39, 0.20)	0.546	0.15 (0.92)	0.10 (0.83)	0.01 (−0.28, 0.30)	0.954
**Lifestyle behaviours**								
Fruit intake grams/day	10.16 (134.37)	24.58 (135.73)	6.47 (−40.05, 52.98)	0.785	4.45 (157.81)	25.37 (122.68)	−11.57 (−58.59, 35.46)	0.630
Vegetable intake grams/day	2.02 (293.56)	10.45 (168.53)	41.09 (−17.65, 99.83)	0.170	34.29 (186.50)	−17.88 (170.20)	51.97 (−5.63, 109.57)	0.077
Dietary Quality Score	0.18 (2.65)	0.11 (2.07)	−0.18 (−0.92, 0.55)	0.627	0.10 (2.37)	0.44 (2.39)	−0.37 (−1.05, 0.30)	0.277

Note: * Change in intervention relative to control adjusted for variable at baseline and cluster size category (Small < 40; Large ≥ 40). Abbreviations: BMI = Body Mass Index; CI = Confidence Interval; HbA1c = Haemoglobin A1c; HDL = High density lipoprotein; LDL = Low density lipoprotein; SD = Standard deviation.

## Data Availability

The datasets generated during and/or analysed during the current study are available from the corresponding author on reasonable request.

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
