# Peer review of "Drivers with and without Obesity Respond Differently to a Multi-Component Health Intervention in Heavy Goods Vehicle Drivers"

_ijerph, 2022, doi:10.3390/ijerph192315546_

Round 1
Reviewer 1 Report
This paper is a good follow on from previous work published on the SHIFT study. It is very unfortunate that COVID hampered efforts to be able to follow up on participants long-term, nevertheless this manuscript greatly contributes to the current gap in research of rigorous evaluation of health intervention aimed at truck drivers.
Comments:
1. Minor spell check in the following lines:
Line 145- fix reference format
Line 166 – Fix error “service25”
Line 191 check grammar and placing of brackets- add in women?
2. Content:
Methods (2.3): Define Light physical activity (give example) and Moderate-to-vigorous physical activity.
Methods (2.4) - is a “cab workout” example of wither LPA or MVPA?
Table 1 – Please define Q-risk and how this was measured/calculated.
Table 1- Please define dietary quality score.
Discussion:
It was interesting that SHIFT program did not seem to result in significant changes in Lifestyle behaviours and mostly impacted exercise/physical activity among participants. BMI reduction/Weight loss etc was mainly attributed to physical activity. Perhaps the researchers could comment on why this might be?
Author Response
Reviewer 1
Point 1:This paper is a good follow on from previous work published on the SHIFT study. It is very unfortunate that COVID hampered efforts to be able to follow up on participants long-term, nevertheless this manuscript greatly contributes to the current gap in research of rigorous evaluation of health intervention aimed at truck drivers.
Response 1: We thank the reviewer for giving up their time to critically review our manuscript, and for the constructive comments provided.
Point 2: Minor spell check in the following lines:
Line 145- fix reference format
Line 166 – Fix error “service25”
Response 2: We thank the reviewer for noticing these mistakes. The reference format has been corrected in both lines.
Point 3:Line 191 check grammar and placing of brackets- add in women?
Response 3: The brackets have been removed and women have been added.
Point 4:
- Content:
Methods (2.3): Define Light physical activity (give example) and Moderate-to-vigorous physical activity.
Response 4: We agree and have added some information on light and moderate-to-vigorous physical activity.
Point 5: Methods (2.4) - is a “cab workout” example of wither LPA or MVPA?
Response 5: We added the information that the cab workout included resistance exercises in light intensities.
Point 6: Table 1 – Please define Q-risk and how this was measured/calculated.
Response 6: We agree and added in the methods section information on the Q-risk.
“The risk of having a cardiovascular event within the next 10 years was calculated using the QRISK2 algorithm (Hippisley-Cox et al., 2008).”
Point 7: Table 1- Please define dietary quality score.
Response 7: We agree and added a definition of how the dietary quality score was measured/ calculated in the methods section.
“Diet was measured using a short form food frequency questionnaire (Cleghorn et al., 2016). A dietary quality score (DQS) was then calculated based on fruit, vegetable, oily fish, fat, and non-milk extrinsic sugar intake. Scores between one to three were given for each category, with a score of three meeting the UK recommendations were met for that component. The maximum achievable DQS was 15.”
Point 8:
Discussion:
It was interesting that SHIFT program did not seem to result in significant changes in Lifestyle behaviours and mostly impacted exercise/physical activity among participants. BMI reduction/Weight loss etc was mainly attributed to physical activity. Perhaps the researchers could comment on why this might be?
Response 8: We feel there may have been a slight misinterpretation here, as we do not intend to attribute the weight loss solely to physical activity, within drivers with obesity. To help clarify this, and to respond to this comment, we have added the following text to lines.
“The increase in steps observed in drivers with obesity at 6-months is unlikely to fully explain the weight loss observed in this group alone, suggesting that some changes in dietary behaviours may have also occurred. Whilst dietary changes were reported by intervention participants within our process evaluation (Guest et al. 2022), the sensitivity of our Food Frequency Questionnaire was not sufficient however to detect changes in dietary behaviours (Clemes et al. 2022).”

Reviewer 2 Report
This paper contributes to the literature about sedentary workers. The manuscript is well prepared and easy to read. I noted extra parentheses in Table 3 (mean (SD)), but did not have substantial comments beyond this.
Author Response
Reviewer 2
Point 1: This paper contributes to the literature about sedentary workers. The manuscript is well prepared and easy to read
Response 1: We thank the reviewer for giving up their time to critically review our manuscript, and for the constructive comment provided.
Point 2: I noted extra parentheses in Table 3 (mean (SD)), but did not have substantial comments beyond this.
Response 2: We thank the reviewer for noticing the extra parentheses. These have been removed throughout all tables.
